# Fourier Minds, Forget Less: Discrete Fourier Transform for Fast and Robust Continual Learning in LLMs

## Abstract

Continual learning (CL) for large language models (LLMs) is challenged by both catastrophic forgetting and efficiency constraints when facing long sequential tasks. While low-rank adaptation in LoRA-based approaches reduces per-task trainable parameters, the cumulative parameter budget grows with stream length and can be substantial. This limits their applicability in lifelong learning scenarios, especially under strict resource constraints. In this work, we explore the potential of the parameter-efficient Sparse Fourier Transform (SFT) in the context of continual learning. Our preliminary experiments reveal that directly applying SFT in CL settings leads to temporal instability and forgetting. Motivated by this finding, we propose Discrete Fourier Continual Learning (DF-CL), which leverages a spectral decomposition strategy to disentangle shared and task-specific knowledge components, facilitating more stable continual learning. By leveraging the orthogonality properties inherent to the SFT bases, DF-CL ensures that task-specific knowledge is encoded within its own dedicated parameter space, minimizing interference between tasks. Furthermore, we introduce a max-magnitude task-weight merging strategy, which enables efficient knowledge consolidation and transfer across sequential tasks. Extensive experiments on both T5-Large and LLaMA2-7B demonstrate the scalability, efficiency, and effectiveness of DF-CL.

## 1 Introduction

Continual learning (CL) aims to enable models to learn a sequence of tasks without revisiting previous data, while maintaining performance across all tasks. A key challenge in CL is catastrophic forgetting, where newly acquired knowledge interferes with previously learned information. To address this, recent CL studies have leveraged large-scale foundation models to enhance transferability and improve performance on streaming tasks. Given the substantial number of parameters in foundation models, these approaches typically incorporate Parameter-Efficient Fine-Tuning (PEFT) strategies such as LoRA (Hu et al., 2022; Liu et al., 2024), adapters (Houlsby et al., 2019; Gao et al., 2024a), and prompt-tuning (Qin & Eisner, 2021; Zhou et al., 2022). By combining CL algorithm design with PEFT techniques, they reduce the computational and memory overhead by fine-tuning only a small subset of parameters, while mitigating forgetting through task-specific adaptation.

Despite these advances, the cost of parameter tuning remains a bottleneck when scaling to long task sequences. In practice, even PEFT-based approaches often require maintaining separate modules for each task, such as task-specific prompts or low-rank adapters. As the number of tasks or the scale of the backbone model increases, these additional components accumulate and lead to substantial memory overhead. For example, the trainable parameters grow significantly when moving from T5 to LLaMA backbones, or when the task number obviously increases, as shown in Figure 1(a). This accumulation not only reduces overall parameter efficiency but also restricts the applicability of such methods in resource-constrained environments, where memory capacity is essential.

To overcome these limitations, we revisit spectral representations and explore the Sparse Fourier Transform (SFT) (Hassanieh et al., 2012) as a compact and expressive alternative. This spectral perspective offers a principled approach to continual learning: low-frequency components can represent stable, general knowledge shared across tasks, while high-frequency components can capture fine-

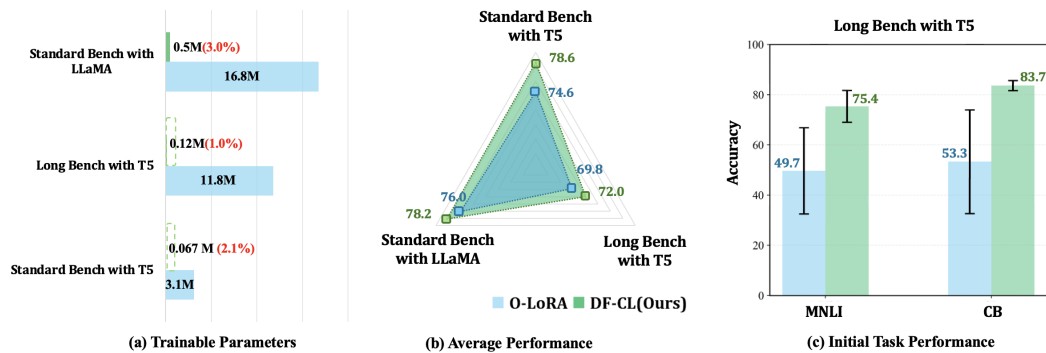

Figure 1: (a) Comparison of trainable parameters: DF-CL updates only 1–3% of total parameters compared to O-LoRA. (b) Performance comparison: DF-CL achieves up to a 4.0% improvement over O-LoRA. (c) Average accuracy and standard deviation on the initial two tasks across sequential training on Long Benchmark (Order 1), showing that DF-CL not only achieves higher accuracy, but also maintains lower variance and thus more stable performance when facing new tasks.

grained, task-specific information. However, our preliminary results show that directly applying SFT to continual learning introduces instability, with noticeable forgetting during task transitions. This is likely due to the lack of explicit constraints across tasks, which makes models prone to overwriting previous knowledge while learning new information, thus exacerbating forgetting (Figure 3(a)).

To address this instability, we introduce Discrete Fourier Continual Learning (DF-CL), a method that **explicitly decouples general and task-specific knowledge in the spectral domain**. We maintain a global set of spectral parameters to represent shared knowledge and learn a small, task-specific set for new information. To further preserve task-specific representations and prevent interference of these subspaces, considering the intrinsic orthogonal nature of fourier bases, we enforce orthogonality among task-specific spectral parameters through coefficient index selection conflict. Moreover, we discover that SFT updates, though compact, can disproportionately affect the model parameters, leading to instability across tasks (Figure 3(b)). Consequently, we propose a max-magnitude **task-weight merging strategy** that selectively integrates the most significant task-specific parameters into the global knowledge base. This merging mechanism effectively balances plasticity and stability, enabling the model to retain prior knowledge while adapting to new tasks. As a result, our **DF-CL combines the parameter efficiency of SFT with strong knowledge retention and task adaptability**, making it well-suited for continual learning. Illustrated in Figure 1, comprehensive evaluations on both T5 and LLaMA models demonstrate that DF-CL effectively preserves task stability and thus consistently outperforms several strong baselines, while requiring only about 1–3% of trainable parameters.

In summary, our key contributions are as follows:

- We are the first to introduce the SFT into the continual learning setting, aiming to further push the limits of parameter reduction while maintaining task performance.
- We design DF-CL, a spectral continual learning method that explicitly decouples general and task-specific knowledge and employs a max-magnitude merging strategy to maintain a balance between knowledge retention and task adaptability.
- Extensive experiments across multiple CL benchmarks demonstrate that DF-CL achieves superior performance while using substantially fewer trainable parameters than O-LoRA.

## 2 RELATED WORK

**Classic Continual Learning.** Traditional continual learning aims to sequentially acquire knowledge from a series of tasks, achieving strong performance on new tasks while retaining previously learned knowledge. These CL methods are generally grouped into three categories: rehearsal-based, regularization-based, and architecture-based approaches. Rehearsal-based methods (Riemer et al., 2018; Chaudhry et al., 2019; Wang et al., 2023a; 2024b) maintain a memory buffer to replay data from previous tasks, alleviating forgetting by directly revisiting old samples. Regularization-based approaches (Kirkpatrick et al., 2017; Li & Hoiem, 2017; Lee et al., 2019; Wu et al., 2024) intro-

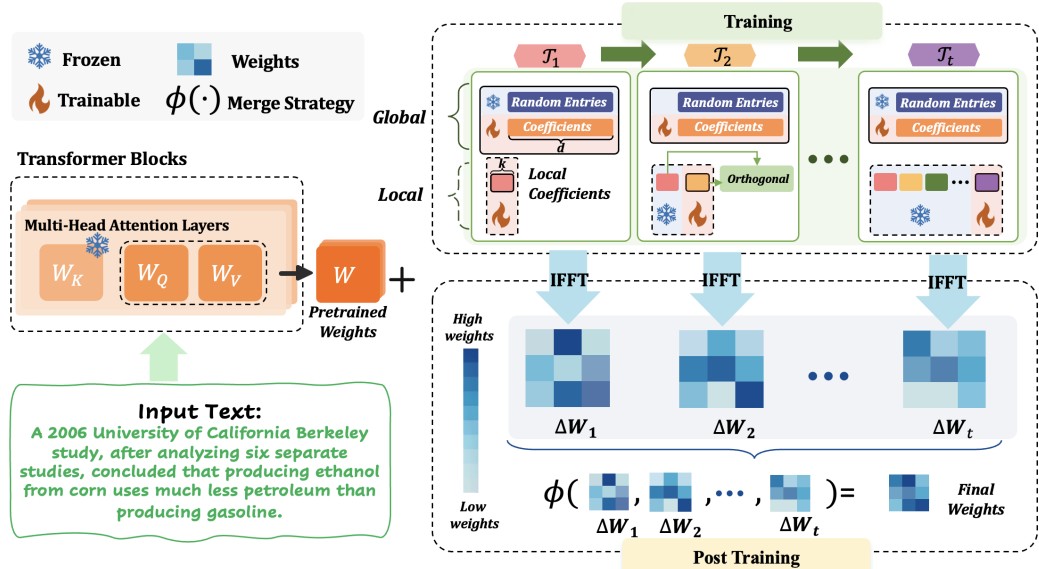

Figure 2: **Overview of the proposed DF-CL framework.** For each pre-trained weight matrix $\mathbf{W}$, we sequentially train a discrete spectral matrix for each task $\mathcal{T}_t$. A shared random spectral entry matrix is initialized and reused across all transformer layers and tasks. DF-CL maintains a global trainable coefficient vector in $\mathbb{R}^d$ shared across all tasks, and a task-specific local coefficient vector in $\mathbb{R}^k$ that is only updated during the current task. The weight updates $\Delta \mathbf{W}$ are obtained by applying the inverse discrete Fourier transform (IDFT) to the updated spectral matrix. After completing all tasks, a task-weight merging strategy $\phi(\cdot)$ is applied to produce the final adapted weights. For all $L$ adapted layers, DF-CL stores only $(d + k \times \mathcal{T}_t) \times L$ parameters, ensuring high parameter efficiency.

duce penalty terms to constrain sensitive parameter updates, thus preserving important knowledge. Architecture-based methods (Mallya et al., 2018; Ebrahimi et al., 2020; Ramesh & Chaudhari, 2021) expand or dynamically modify the model architecture to accommodate new tasks without interfering with existing representations. While prior CL methods effectively reduce forgetting, they are seldom combined with large foundation models, limiting their scalability. We propose DF-CL, which leverages large foundation models and employs the Discrete Fourier Transform (Xu et al., 2020; Gao et al., 2024b) to reduce trainable parameters, improving both efficiency and practicality for CL.

**CL with Foundation Models.** Recent CL works hope to leverage large foundation models to improve performance on sequential tasks. These methods typically adopt Parameter-Efficient Fine-Tuning (PEFT) techniques to adapt models effectively while mitigating forgetting. A key challenge is improving efficiency without string a large number of task-specific parameters, which is particularly important for maintaining stability in long sequential tasks Wu et al. (2025). For example, LoRA-based methods, such as O-LoRA (Wang et al., 2023b) and MO-CL (Wang et al., 2024a), enhance training efficiency by applying low-rank adaptation for task-specific tuning and incorporate various mechanisms to alleviate forgetting. While Prompt-based methods, like L2P (Wang et al., 2022c), DualPrompt (Wang et al., 2022a), and CODA-Prompt (Smith et al., 2023), introduce lightweight learnable prompts as task-specific knowledge to mitigate forgetting and improve training efficiency. In contrast to these approaches based on widely used PEFT techniques, we propose a novel method that leverages the Inverse Discrete Fourier Transform to further reduce trainable parameters more significantly. By explicitly isolating task-specific and general knowledge and adopting a merging strategy, our method further ensures stable CL performance.

**Sparse Fourier Transform.** Sparse Fourier Transform (SFT) has been introduced into deep learning to leverage sparse spectral coefficients for representation learning (Rawat et al., 2019; Ehrlich & Davis, 2019; Xu et al., 2020). Previous studies (Yang & Xie, 2016; Chen & Chi, 2013) have shown that SFT can effectively reconstruct data with extremely few parameters, even when the underlying signals are not strictly frequency-sparse. Building on these works, FourierFT (Gao et al., 2024b) applies SFT to parameter-efficient fine-tuning by modeling the weight update as a spatial-domain matrix and learning its sparse spectral coefficients. In this work, we extend FourierFT to

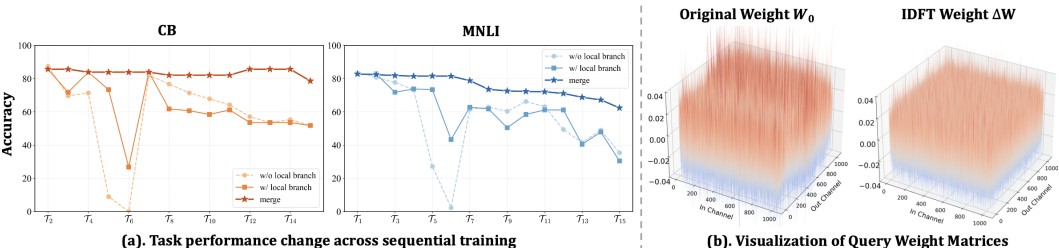

Figure 3: (a). Direct application of FourierFT leads to temporal forgetting. Adding local branches mitigates this issue, while task-weight merging further stabilizes previous task performance. (b). Perturbing only 0.5% of spectral coefficients yields an IDFT weight $\Delta \mathbf{W}$ comparable in scale to $\mathbf{W}_0$, highlighting small spectral perturbations can induce large shifts in the weight domain.

the continual learning setting and propose DF-CL, a novel framework that incorporates orthogonal task-specific Fourier branches and a task-aware weight merging strategy. DF-CL effectively mitigates the transient forgetting issues observed in the original FourierFT and consistently improves overall performance across sequential tasks.

## 3 METHOD

### 3.1 PRELIMINARIES

**Problem Formulation.** Continual learning trains predictive model $f_\theta(\cdot)$ on a sequence of $N$ tasks $\{\mathcal{T}_1, \mathcal{T}_2, \ldots, \mathcal{T}_N\}$, where each task $\mathcal{T}_t$ is associated with a dataset $\mathcal{D}_t = \{(\mathbf{x}_i^{(t)}, y_i^{(t)})\}_{i=1}^{|\mathcal{D}_t|}$ containing $|\mathcal{D}_t|$ labeled samples. Under the common CL setting, past task data are unavailable during training, and the objective for the current task $\mathcal{T}_t$ is:

$$\mathcal{L}_f = - \sum_{(\mathbf{x}, y) \in \mathcal{D}_t} \log f_\theta(y \mid \mathbf{x}). \tag{1}$$

### 3.2 DISCRETE FOURIER CL

**Sparse Fourier Transformation**. To further explore how much we reduce training parameters without sacrificing performance, we draw inspiration from FourierFT (Gao et al., 2024b), which introduces a sparse spectral entry matrix to significantly reduce parameter overhead. Leveraging its compactness and expressiveness, we extend FourierFT to CL tasks. Specifically, to update a weight matrix $\mathbf{W} \in \mathbb{R}^{m \times n}$, we randomly initialize a *spectral entry matrix* $\mathbf{M} \in \mathbb{R}^{2 \times d}$, where each column defines a discrete 2D frequency coordinate. A corresponding coefficient vector $\mathbf{x} \in \mathbb{R}^d$ is initialized from a standard Gaussian distribution. The sparse *spectral matrix* $\mathbf{N} \in \mathbb{R}^{m \times n}$ is constructed as:

$$\mathbf{N}_{u,v} = \begin{cases} x_l & \text{if } u = \mathbf{M}_{0,l} \wedge v = \mathbf{M}_{1,l}, \\ 0 & \text{otherwise.} \end{cases} \tag{2}$$

Then, the *spatial matrix* $\mathbf{S} \in \mathbb{C}^{m \times n}$ is recovered using the inverse 2D Discrete Fourier Transform (IDFT):

$$\mathbf{S}_{p,q} = \mathcal{F}^{-1}(\mathbf{N})_{p,q} = \sum_{u=0}^{m-1} \sum_{v=0}^{n-1} \mathbf{N}_{u,v} \cdot e^{i2\pi\left(\frac{pu}{m} + \frac{qv}{n}\right)}, \tag{3}$$

where $\mathcal{F}^{-1}(\cdot)$ denotes the inverse fourier transform. The final $\Delta \mathbf{W}$ is obtained by taking the real part of the spatial matrix, scaled by a stable scalar $\beta$:

$$\mathbf{W} = \mathbf{W}_0 + \beta \cdot \Delta \mathbf{W} = \mathbf{W}_0 + \beta \cdot \Re(\mathbf{S}). \tag{4}$$

By sharing the same spectral indices $\mathbf{M}$, it achieves substantial parameter savings compared to LoRA or prompt-based methods. For a LLM with $L$ layers, this reduces the total number of trainable parameters to $d \times L$, where $d$ is the number of selected frequency entries.

**DF-CL.** While the Sparse Fourier Transform offers the advantage of significantly reducing the number of trainable parameters, its applicability to CL has not yet been explored. To bridge this gap, we extend the SFT framework to the CL setting. However, a straightforward application fails to address the problem of forgetting and suffers from the stability gap problem, as shown in Figure 3 (a). The stability gap refers to the phenomenon where a model experiences severe temporary forgetting during the CL process (De Lange et al., 2022). For example, in a preliminary experiment, the performance on the CB dataset dropped drastically from 71.43 to 8.93 immediately after training on task five (QPP). Although the final accuracy on CB recovered to 51.79 after completing the full task sequence, such temporary degradation is unacceptable in real-world applications.

(*Task-specific Branch.*) To overcome this limitation, we first hypothesize that the observed stability gap stems from the use of a shared coefficient vector $\mathbf{x}$ across all tasks. While parameter-efficient, this design neglects task-specific knowledge and is prone to forgetting when the data distribution shifts significantly between tasks. To address this limitation, we introduce a **task-specific coefficient vector** $\mathbf{x}_t \in \mathbb{R}^k$ for each new task $\mathcal{T}_t$. This enables the construction of a task-specific spectral matrix $\mathbf{N}_t$. Meanwhile, the **global spectral matrix** $\mathbf{N}_{\text{global}}$ serves as a shared base across tasks, while during task $\mathcal{T}_t$, we jointly optimize the shared coefficients $\mathbf{x}_{\text{global}}$ and the task-specific vector $\mathbf{x}_t$. Formally, the overall weight update at task $\mathcal{T}_t$ is given by:

$$\mathbf{W} = \mathbf{W}_0 + \beta \cdot \Delta\mathbf{W}^{(t)} = \mathbf{W}_0 + \beta \cdot \Re(\mathcal{F}^{-1}(\mathbf{N}^{(t)}))$$

$$= \mathbf{W}_0 + \beta \cdot \Re(\mathcal{F}^{-1}(\mathbf{N}_{\text{global}} + \sum_{i=1}^{t} \mathbf{N}_i)). \tag{5}$$

The coefficients from previous tasks, $\mathbf{x}_1, \ldots, \mathbf{x}_{t-1}$, are kept frozen to mitigate knowledge forgetting, and only $\mathbf{x}_{\text{global}}$ and $\mathbf{x}_t$ will be updated. Notably, we constrain the newly introduced coefficients to be associated with *non-overlapping indices* in the spectral matrix $\mathbf{N}_{\text{global}}$ and $\{\mathbf{N}_i\}_{i=1}^{t-1}$. This design induces *orthogonal subspaces* for each task-specific branch—thanks to the inherent orthogonality of Fourier bases—thereby avoiding interference. This formulation allows the model to incrementally expand its representational capacity while maintaining knowledge from earlier tasks, making the model particularly well-suited to continual-learning scenarios where tasks may differ significantly. In practice, we will choose a small task-specific dimensionality $k<d$, ensuring that the additional parameter cost per task remains minimal. As shown in Figure 3(a), adding a lightweight task-specific branch alleviates the stability gap, but forgetting remains, motivating our task-weight merging.

(*Task-weight Merging*). To investigate the source of the instability, we further hypothesize that the remained forgetting observed in Figure 3(a) stems from the sensitivity of model weights to updates in the spectral domain. To verify this, we randomly initialize a sparse spectral entry matrix $\mathbf{M}$ and transform it back into the spatial domain. As presented in Figure 3(b), the reconstructed $\Delta\mathbf{W}$ exhibits a distribution and scale comparable to that of the original weight matrix $\mathbf{W}_0$. This observation suggests that even small perturbations in the spectral domain can propagate into disproportionately large and unstable changes in the model weights. To address this, we draw inspiration from multi-task learning and incorporate *model merging* techniques (Marczak et al., 2024) during the post-training phase. Model merging facilitates effective knowledge consolidation by combining independently trained memory components from different tasks using tailored strategies. In our case, we focus on merging the real-valued spatial matrices $\Delta\mathbf{W}^{(i)}$ obtained from sequential task training ($i = 1, 2, \ldots, t$). We empirically evaluate several merging strategies, such as element-wise **mean** and **max**, and observe that minimizing large parameter shifts consistently yields better performance. Specifically, for each element with the coordinate of $(p, q)$ in weight matrix, we perform magnitude-based merging across tasks to obtain:

$$\Phi_{p,q} = \arg \max_{i \in \{1, \ldots, T\}} \left| \Delta\mathbf{W}_{p,q}^{(i)} \right|, \tag{6}$$

$$\Delta\mathbf{W}_{p,q} = \Delta\mathbf{W}_{p,q}^{(\Phi_{p,q})}. \tag{7}$$

Finally, the merged model is computed as:

$$\mathbf{W}_{\text{final}} = \mathbf{W}_0 + \beta \cdot \Delta\mathbf{W}. \tag{8}$$

During inference, the merged spectral weights can be directly incorporated into the base model without introducing any additional overhead.

---

**Algorithm 1** DF-CL Algorithm

---

**Input:** Pretrained weight matrix from foundation model $\mathbf{W}_0 \in \mathbb{R}^{m \times n}$, training datasets for tasks $\{\mathcal{D}_1, \ldots, \mathcal{D}_N\}$, hyper-parameters $d, k$.
**Output:** Updated weight matrix $\mathbf{W}_{\text{final}}$

1: Initialize coefficient vector $\mathbf{x}_{\text{global}} \in \mathbb{R}^d$
2: Sample $d$ coordinate pairs $\mathcal{M}_{\text{global}} = \{(\mathbf{M}_{1,i}, \mathbf{M}_{2,i})\}_{i=1}^d$ from set $[m] \times [n]$ without replacement
3: Obtain spectral matrix $\mathbf{N}_{\text{global}}$ based on Eqn. (2)
4: **for** $t = 1$ to $N$ **do**
5:     Initialize new task-specific coefficient vector $\mathbf{x}_t \in \mathbb{R}^k$
6:     Freeze previous task-specific coefficients $\{\mathbf{x}_i\}_{i=1}^{t-1}$; unfreeze $\mathbf{x}_t$ and $\mathbf{x}_{\text{global}}$
7:     Sample $k$ coordinate pairs $\mathcal{M}_t$ from set $([m] \times [n]) \setminus \left( \mathcal{M}_{\text{global}} \cup \bigcup_{i=1}^{t-1} \mathcal{M}_i \right)$ without replacement
8:     Obtain spectral matrix $\mathbf{N}_t$ with $\mathcal{M}_t$ based on Eqn. (2)
9:     Compute cross-entropy loss on $\mathcal{D}_t$ using $\mathbf{W}$ based on Eqn. (5), and update unfrozen coefficients with mini-batches and AdamW optimizer
10:     Save all coefficients $\mathbf{x}$ and $\mathcal{M}$ in this round to compute $\mathbf{N}^{(t)} = \mathbf{N}_{\text{global}} + \sum_{i=1}^t \mathbf{N}_i$
11: **end for**
12: Compute $\mathbf{W}_{\text{final}}$ based on Eqns. (6)–(8), where $\Delta\mathbf{W}^{(i)} = \Re(\mathcal{F}^{-1}(\mathbf{N}^{(i)}))$ for each task $\mathcal{T}_i$
13: **return** $\mathbf{W}_{\text{final}}$

---

**Parameter Analysis.** We provide a theoretical comparison of the trainable parameter count between O-LoRA (Wang et al., 2023b) and our DF-CL, as summarized in Table 1. O-LoRA introduces a pair of trainable matrices, $\mathbf{A}$ and $\mathbf{B}$, for each tunable module. Assuming a total of $\mathcal{T}_t$ tasks and $L$ trainable modules, the number of parameters is:

$$\mathcal{N}_{\text{O-LoRA}} = 2 \times \dim \times L \times r \times \mathcal{T}_t, \tag{9}$$

where $r$ is the LoRA rank and $\dim = m = n$ is the dimensionality of each module. In contrast, DF-CL utilizes $d$ global coefficients and $k$ task-specific coefficients per task. The total number of trainable parameters is:

$$\mathcal{N}_{\text{DF-CL}} = (d + k \times \mathcal{T}_t) \times L. \tag{10}$$

For example, when fine-tuning the query and value matrices of T5-Large (24 layers $\times$ 2 modules, so $L = 48$), and assuming $\mathcal{T}_t = 15$, we have:

- O-LoRA: $\mathcal{N}_{\text{O-LoRA}} = 11.8\text{M}$ parameters with $r = 8$ and $\dim = 1024$,
- DF-CL: $\mathcal{N}_{\text{DF-CL}} = 120\text{k}$ parameters with $d = 1000$ and $k = 100$.

This reduces the trainable parameter count to approximately **1.1%** compared to O-LoRA. Moreover, the parameter efficiency advantage of DF-CL becomes increasingly significant as either the number of tasks or the model size grows, illustrated in Table 1.

Table 1: Trainable parameter comparison between O-LoRA and DF-CL. We suppose the trainable modules are query and value weight matrices across each Transformer layer.

| LLMs | O-LoRA | | | DF-CL | | | |
|---|---|---|---|---|---|---|---|
| | # Tasks | $r$ | #Params | # Tasks | $d$ | $k$ | # Params |
| T5-Large | 4 | 8 | 3.1M | 4 | 1000 | 100 | 67.2k (↓ 97.8%) |
| | 15 | 8 | 11.8M | 15 | 1000 | 100 | 120k (↓ 98.9%) |
| | 4 | 16 | 6.3M | 4 | 1000 | 200 | 86.4k (↓ 98.6%) |
| | 15 | 16 | 23.6M | 15 | 1000 | 200 | 192k (↓ 99.2%) |
| LLaMA2-7B | 4 | 8 | 16.8M | 4 | 1000 | 250 | 128k (↓ 99.2%) |
| | 15 | 8 | 62.9M | 15 | 1000 | 250 | 304k (↓ 99.5%) |
| | 4 | 16 | 33.6M | 4 | 1000 | 500 | 192k (↓ 99.4%) |
| | 15 | 16 | 125.8M | 15 | 1000 | 500 | 544k (↓ 99.6%) |

## 4 EXPERIMENTS

### 4.1 EXPERIMENTAL SETUP

**Datasets.** Following O-LoRA (Wang et al., 2023b), we evaluate our DF-CL on two widely used benchmarks, the Standard benchmark and Long benchmark. Standard benchmark (Zhang et al.,

Table 2: Experimental results on two CL benchmarks with T5-Large(Raffel et al., 2020). **Params.** represents the parameter usage for training.

| Method | Standard ($N = 4$) | | | | | Long ($N = 15$) | | | | |
|---|---|---|---|---|---|---|---|---|---|---|
| | Params. | Order1 | Order2 | Order3 | avg | Params. | Long1 | Long2 | Long3 | avg |
| PerTaskFT | - | 70.0 | 70.0 | 70.0 | 70.0 | - | 78.1 | 78.1 | 78.1 | 78.1 |
| MTL | - | 80.0 | 80.0 | 80.0 | 80.0 | - | 76.5 | 76.5 | 76.5 | 76.5 |
| Zero-Shot | - | 0.0 | 0.0 | 0.0 | 0.0 | - | 13.4 | 13.4 | 13.4 | 13.4 |
| SeqLoRA | 0.8M | 25.7 | 24.0 | 35.2 | 28.3 | 3.0M | 12.3 | 10.1 | 10.1 | 10.8 |
| Replay Memory | 770M | 55.2 | 56.9 | 61.3 | 57.8 | 770M | 55.0 | 54.6 | 53.1 | 54.2 |
| EWC | 770M | 48.7 | 47.7 | 54.5 | 50.3 | 770M | 45.3 | 44.5 | 45.6 | 45.1 |
| LwF | 4.1M | 54.4 | 53.1 | 49.6 | 52.3 | 15.7M | 50.1 | 43.1 | 47.4 | 46.9 |
| L2P | 96.0k | 60.3 | 61.7 | 61.1 | 60.7 | 0.4M | 57.5 | 53.8 | 56.9 | 56.1 |
| LFPT5 | 1.2M | 67.6 | 72.6 | 77.9 | 72.7 | 4.6M | 70.4 | 68.2 | 69.1 | 69.2 |
| IncLoRA | 3.1M | 68.6 | 59.7 | 75.0 | 67.8 | 11.8M | 60.3 | 60.5 | 53.2 | 58.0 |
| MIGU | 3.1M | 77.2 | 76.7 | 75.4 | 76.4 | 11.8M | 71.3 | 67.7 | 67.3 | 68.7 |
| O-LoRA | 3.1M | 74.9 | 73.4 | 75.6 | 74.6 | 11.8M | 71.5 | 66.7 | 71.3 | 69.8 |
| LB-CL | 3.2M | 76.9 | 76.5 | 76.8 | 76.7 | 11.8M | 68.4 | 67.3 | 71.8 | 69.2 |
| MoCL | 3.3M | 75.6 | 75.4 | 76.7 | 75.9 | - | - | - | - | - |
| **DF-CL** | **67.2k** | **78.7** | **78.7** | **78.4** | **78.6** | **120.0k** | **72.3** | **70.9** | **73.2** | **72.1** |

2016) consists of four text classification datasets, including Amazon reviews, Yelp reviews, DBpedia, and Yahoo Answers. Long benchmark further involve AG News classification task, four datasets from GLUE (Wang et al., 2018), five tasks from SuperGLUE (Wang et al., 2019), and IMDB movie reviews dataset (Maas et al., 2011). More details are listed in Appendix B.

**Implementation Details.** We implement our DF-CL framework on both an encoder-decoder architecture (T5-Large (Raffel et al., 2020)) and a decoder-only model (LLaMA2-7B (Touvron et al., 2023)). Specifically, DF-CL is applied only to the query and value matrices across all transformer layers, ensuring training efficiency. For global parameters, we allow $d = 1000$ trainable spectral coefficients out of the full spectral space, i.e., $1024^2$ for T5-Large and $4096^2$ for LLaMA2-7B. These global coefficients are shared and updated throughout all tasks in the continual learning sequence. For task-specific adaptation, we allocate $k = 100$ coefficients per task on T5-Large and $k = 500$ on LLaMA2-7B. These coefficients are independently updated for each task and remain frozen afterward, enabling task-specialized learning while preserving knowledge from previous tasks. Additional implementation details and hyperparameter settings can be found in Appendix B.

**Metrics.** The $a_{i,j}$ denotes the test accuracy on the $i$-th task $\mathcal{T}_i$ after training on $\mathcal{T}_j$. We adopt the Average Accuracy as the main evaluation metric, which is calculated as the mean accuracy across all seen tasks,

$$\text{Acc} = \frac{\sum_{i=1}^{N} |\mathcal{D}_i| \cdot a_{i,N}}{\sum_{i=1}^{N} |\mathcal{D}_i|},$$

where $|\mathcal{D}_i|$ represents the number of test samples in task $\mathcal{T}_i$.

**Baselines.** We compare DF-CL with state-of-the-art baselines: including PerTaskFT, MTL, Zeroshot, SeqLoRA, Replay Memory, EWC (Kirkpatrick et al., 2017), LwF (Li & Hoiem, 2017), L2P (Wang et al., 2022b), LFPT5 (Huang et al., 2021), IncLoRA, MIGU (Du et al., 2024), O-LoRA (Wang et al., 2023b) and LB-CL (Qiao & Mahdavi, 2024). Details for each method are listed in Appendix C.

## 4.2 MAIN RESULTS

**DF-CL Performs Well on Different LLM Backbones.** We present the experimental results on both T5-Large and LLaMA2-7B in Table 2 and Table 3, respectively. Our proposed DF-CL consistently achieves superior performance with significantly fewer trainable parameters compared to LoRA-based methods. For instance, on the T5-Large model, DF-CL surpasses O-LoRA by 4.0% and LB-CL by 1.9% on the Standard benchmark, while utilizing only approximately 2% of their parameters. Although prompt-based approaches like L2P are highly parameter-efficient, they exhibit

Table 3: Experimental results on Standard CL benchmarks with LLaMA2-7b (Touvron et al., 2023).

| Method | | **Standard** ($N = 4$) | | | |
|--------|---------|--------|--------|--------|------|
| | Params. | Order1 | Order2 | Order3 | avg |
| PerTaskFT | - | 79.9 | 79.9 | 79.9 | 79.9 |
| MTL | - | 80.3 | 80.3 | 80.3 | 80.3 |
| Zero-shot | - | 0.0 | 0.0 | 0.0 | 0.0 |
| SeqLoRA | 4.2M | 73.4 | 75.6 | 75.5 | 74.8 |
| IncLoRA | 16.8M | 75.9 | 72.6 | 76.8 | 75.1 |
| O-LoRA | 16.8M | 76.8 | 75.7 | 75.7 | 76.0 |
| MoCL | 19.7M | **78.4** | 77.7 | 78.4 | 78.2 |
| **DF-CL** | **0.5M** | **78.4** | **77.9** | **78.9** | **78.4** |

Table 4: Ablation of coefficient number $(d, k)$ on LLaMA2-7B.

| d | k | Order1 |
|------|-----|--------|
| 500 | 0 | 66.4 |
| 1000 | 0 | 74.7 |
| 2000 | 0 | 76.9 |
| 1000 | 100 | 76.5 |
| 1000 | 250 | 77.4 |
| 1000 | 500 | 77.8 |

Figure 4: Test Acc. throughout continual training on T5.

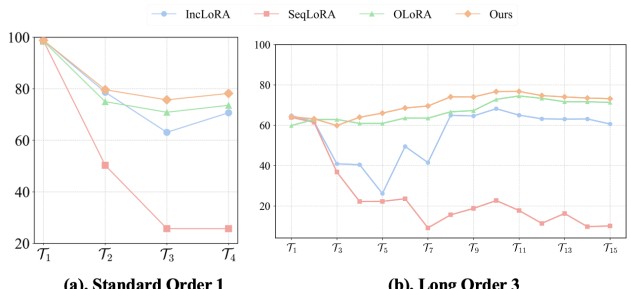

(a). Standard Order 1    (b). Long Order 3

substantially lower performance. For example, L2P underperforms by more than 17.9% compared to DF-CL. Moreover, DF-CL also demonstrates competitive performance on the LLaMA2-7B model, showcasing strong generalization across different large language model backbones. Notably, DF-CL tunes only 1% of the parameters compared to LoRA-based baselines on LLaMA2-7B—for instance, using just 0.2M parameters, DF-CL achieves comparable results to MoCL with 19.7M parameters. These results further underscore the effectiveness and scalability of DF-CL with different LLMs.

**DF-CL Keeps Stability across Varied Task Length.**    To assess the scalability of DF-CL on longer task sequences, we evaluate it against several state-of-the-art baselines on the Long Benchmark, which includes 15 diverse NLP tasks. As shown in Table 2, DF-CL consistently outperforms or matches the performance of leading methods across different task orders, demonstrating its strong generalization ability in continual learning. Notably, as the number of tasks increases, the parameter count for LoRA-based methods grows significantly. In contrast, DF-CL maintains high parameter efficiency, achieving competitive results with only 1% of the parameter usage (0.12M vs. 11.8M). These findings highlight the effectiveness of DF-CL for resource-constrained continual learning scenarios.

**Stability across Tasks.**    To demonstrate the stability of DF-CL throughout the continual learning process, we plot the test accuracy after completing each task across benchmarks with varying task sequence lengths (see Figure 4). Compared with other baselines, DF-CL exhibits notably smaller performance fluctuations and consistently maintains high accuracy throughout training. This stability underscores DF-CL's strong resistance to catastrophic forgetting and its ability to adapt to new tasks without compromising previously acquired knowledge. These results validate the effectiveness of the proposed task-specific branches in mitigating stability issues, as discussed in Section 3.2. Such robustness is particularly valuable in more demanding scenarios like the Long benchmark, where the number of sequential tasks is significantly larger.

### 4.3 ABLATION STUDIES

**Impact of DF-CL Components.**    To better understand the contribution of each component in our proposed DF-CL, we perform ablation studies on T5-Large model from two key perspectives: (1)

Table 5: Ablation study of different components of DF-CL on T5-Large model.

| Local Branch | Merging | Standard ($N = 4$) | | | | Long ($N = 15$) | | | |
|:---:|:---:|:---:|:---:|:---:|:---:|:---:|:---:|:---:|:---:|
| | | Order1 | Order2 | Order3 | avg | Long1 | Long2 | Long3 | avg |
| ✗ | ✓ | 77.9 | 77.3 | 77.2 | 77.5 | 70.0 | 70.2 | 67.4 | 69.2 |
| ✓ | ✗ | 77.7 | 78.2 | 77.3 | 77.7 | 70.9 | 69.4 | 70.0 | 69.7 |
| ✓ | ✓ | **78.7** | **78.7** | **78.4** | **78.6** | **72.3** | **70.9** | **73.2** | **72.1** |

*Task-specific Branches*, which introduce orthogonal and task-related coefficient vectors to alleviate interference; and (2) *Task-weight Merging*, which facilitates knowledge reuse by integrating parameters from previously learned tasks. As shown in Table 5, removing either component leads to a significant drop in performance, confirming their effectiveness in mitigating forgetting and enhancing learning stability. In particular, removing the task-specific branches causes a notable decline in accuracy, indicating that the orthogonal coefficient vectors are critical for maintaining task independence and reducing conflict between old and new knowledge. Furthermore, replacing our task-weight merging strategy with a naive addition operation yields inferior results, suggesting that effective consolidation requires careful selection of parameter updates to avoid forgetting.

**Impact of Merging Strategies.** We present the ablation results of the mean merging and max-magnitude merging strategies in Table 6. It is evident that max-magnitude merging consistently outperforms mean merging, and is therefore adopted as the final strategy in DF-CL. Surprisingly, the mean averag-

Table 6: Ablation of max and mean merging strategies.

| T5-Large | Order 1 | Order 2 | Order3 |
|:---|:---:|:---:|:---:|
| w/o Merge | 77.7 | 78.2 | 77.3 |
| Mean Merge | 76.8 | 64.1 | 77.8 |
| Max Merge | **78.7** | **78.7** | **78.4** |

ing strategy leads to performance degradation on both Order1 and Order2 evaluations, highlighting the importance of selecting an appropriate merging direction. We attribute this to the assumption that parameters with larger magnitudes tend to carry greater importance, as similarly discussed in MAGMAX (Marczak et al., 2024). Consequently, DF-CL achieves further performance gains by preserving high-magnitude parameters. These findings highlight the necessity of our task-weight merging strategy, which not only stabilizes training but also leads to consistent performance improvements across tasks.

**Ablation of coefficient number.** The number of coefficients in the global branch ($d$) and the local branch ($k$) plays a critical role in balancing downstream performance and training cost. In our DF-CL framework, the total number of trainable parameters is given by $(d + k \times \mathcal{T}_t) \times L$. To study the effect of $d$ and $k$, we conduct ablation experiments on LLaMA2-7B using the Standard Benchmarks, without applying the model merging strategy to isolate their influence. As shown in Table 4, increasing $d$ from 500 to 2000 consistently improves performance, demonstrating the scalability of DF-CL. Similarly, increasing $k$ also yields performance gains. We adopt $d = 1000$ and $k = 500$ as the default setting for LLaMA2-7B, balancing performance and efficiency. We anticipate that using larger values (e.g., $d = 2000$) would lead to even stronger results than those reported in Table 3.

## 5 CONCLUSION

In this paper, we propose DF-CL, a Discrete Fourier Transform-based framework for efficient and robust continual learning. Our approach introduces shared global spectral coefficients across tasks and task-specific branches for individual optimization, ensuring orthogonal parameter spaces while mitigating knowledge forgetting through globally shared parameters. To further enhance performance, we develop a task-weight merging strategy that consolidates knowledge from past tasks effectively. Extensive experiments demonstrate that DF-CL consistently outperforms LoRA-based methods across multiple large language model backbones and benchmarks, while requiring significantly fewer trainable parameters.

ETHICS STATEMENT

This research follows the ICLR Code of Ethics. Our work introduces the parameter-efficient Sparse Fourier Transform into the continual learning setting and designs a spectral continual learning method to maintain knowledge retention. All experiments are conducted on publicly available benchmark datasets that do not involve personal or sensitive information. The research does not pose direct ethical or societal risks.

REPRODUCIBILITY STATEMENT

To support reproducibility, we provide a complete description of our proposed DF-CL algorithm in Algorithm 1. The training and evaluation setups are detailed in Section 4.1 and Appendix B. Dataset details are described in Appendix B.

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

APPENDIX

## A  USE OF LLMs

We used ChatGPT5 solely to improve the fluency and clarity of writing. All content was written by the authors, and the model served only as a language assistant. It should not be considered a contributor to the work.

## B  EXPERIMENTAL SETTINGS

| Dataset name | Category | Task | Domain |
|---|---|---|---|
| 1. Yelp | CL Benchmark | sentiment analysis | Yelp reviews |
| 2. Amazon | CL Benchmark | sentiment analysis | Amazon reviews |
| 3. DBpedia | CL Benchmark | topic classification | Wikipedia |
| 4. Yahoo | CL Benchmark | topic classification | Yahoo Q&A |
| 5. AG News | CL Benchmark | topic classification | news |
| 6. MNLI | GLUE | NLI | various |
| 7. QQP | GLUE | paragraph detection | Quora |
| 8. RTE | GLUE | NLI | news, Wikipedia |
| 9. SST-2 | GLUE | sentiment analysis | movie reviews |
| 10. WiC | SuperGLUE | word sense disambiguation | lexical databases |
| 11. CB | SuperGLUE | NLI | various |
| 12. COPA | SuperGLUE | QA | blogs, encyclopedia |
| 13. BoolQA | SuperGLUE | boolean QA | Wikipedia |
| 14. MultiRC | SuperGLUE | QA | various |
| 15. IMDB | SuperGLUE | sentiment analysis | movie reviews |

Table B1: The details of 15 datasets in our continual learning experiments. Among them, NLI refers to natural language inference tasks, while QA denotes question answering. The first five tasks correspond to the standard continual learning benchmark, whereas the remaining ones are used in our long-sequence evaluation.

**Datasets.**  Table B1 summarizes the detailed statistics of the 15 datasets used in our continual learning (CL) experiments, along with their evaluation metrics. The selection includes datasets from well-established benchmarks, including the standard CL benchmark (Zhang et al., 2016), GLUE (Wang et al., 2018), and SuperGLUE (Wang et al., 2019), along with the addition of the IMDB movie reviews dataset (Maas et al., 2011).

**Task Sequence Orders.**  Table B2 summarizes the task sequences used in our continual learning experiments for both T5 and LLaMA models. Orders 1–3 follow the standard benchmarks commonly adopted in previous studies, while Long 1–3 are extended sequences consisting of 15 tasks, following the setup of O-LoRA Wang et al. (2023b).

**Instruction Prompts.**  The prompt templates corresponding to each task type are listed in Table B3. Among them, MNLI, RTE, and CB fall under natural language inference (NLI); Amazon, Yelp, SST-2, and IMDB are grouped as sentiment classification (SC) tasks; while AG News, DBpedia, and Yahoo Answers belong to topic classification (TC).

**Implementation Details.**  All experiments are conducted on NVIDIA RTX 4090 and A100 GPUs, using the DeepSpeed framework for efficient training. DF-CL is implemented based on the O-LoRA training framework, which is released under the MIT license. To ensure reproducibility, we provide the complete source code and training scripts in the supplementary material.

**Training Hyper-parameters.**  For all experiments, we train the models for one epoch per task. We search the weight decay rate in $\{0, 1e-4, 3e-4, 8e-4\}$ and tune the learning rate within the

| Order | Task Sequence |
|-------|---------------|
| order1 | dbpedia → amazon → yahoo → ag |
| order2 | dbpedia → amazon → ag → yahoo |
| order3 | yahoo → amazon → ag → dbpedia |
| long1 | mnli → cb → wic → copa → qqp → boolqa → rte → imdb → yelp → amazon → sst-2 → dbpedia → ag → multirc → yahoo |
| long2 | multirc → boolqa → wic → mnli → cb → copa → qqp → rte → imdb → sst-2 → dbpedia → ag → yelp → amazon → yahoo |
| long3 | yelp → amazon → mnli → cb → copa → qqp → rte → imdb → sst-2 → dbpedia → ag → yahoo → multirc → boolqa → wic |

Table B2: Six different orders of task sequences adopted in our continual learning experiments.

| Task | Prompts |
|------|---------|
| NLI | What is the logical relationship between the "sentence 1" and the "sentence 2"? Choose one from the option. |
| QQP | Whether the "first sentence" and the "second sentence" have the same meaning? Choose one from the option. |
| SC | What is the sentiment of the following paragraph? Choose one from the option. |
| TC | What is the topic of the following paragraph? Choose one from the option. |
| BoolQA | According to the following passage, is the question true or false? Choose one from the option. |
| MultiRC | According to the following passage and question, is the candidate answer true or false? Choose one from the option. |
| WiC | Given a word and two sentences, whether the word is used with the same sense in both sentence? Choose one from the option. |

Table B3: Instructions for different tasks.

range of $[2e-2, 5e-1]$. We set the number of global spectral coefficients to $d = 1000$ for all models. The number of task-specific coefficients is set to $k = 100$ for T5 models and $k = 500$ for LLaMA2-7B models. We use a global batch size of 8 with a gradient accumulation step of 4. For sequence processing, we set the maximum source length to 512, the maximum target length to 50, and the maximum generation length to 50 during both training and evaluation. For more details on hyperparameters and training configurations, please refer to the provided bash scripts in the source code. All experiments are conducted with a fixed seed and reported based on a single run.

## C  BASELINES

We illustrate the baseline methods in our experiments as follow:

- **PerTaskFT**: Trains one independent LoRA module per task from scratch without sharing, serving as a task-isolated upper bound for forgetting avoidance.

- **MTL**: Jointly trains on all task datasets in a multi-task learning setup, representing the ideal performance upper bound with full access to all tasks.

- **Zero-shot**: Evaluates the pretrained model directly on downstream benchmarks without any fine-tuning.

- **SeqLoRA**: Applies a single shared LoRA module across all tasks, updated sequentially as new tasks arrive.

- **Replay Memory**: Maintains a fixed-size memory buffer containing examples from prior tasks, and interleaves them with current task data during training to alleviate forgetting.

- **EWC** (Kirkpatrick et al., 2017): Employs a Fisher Information Matrix-based regularization to preserve important weights by penalizing deviations from previously learned parameters.

- **LwF** (Li & Hoiem, 2017): Prevents forgetting by aligning the current model's predictions with those from prior tasks using a distillation loss, eliminating the need to store old samples.

- **L2P** (Wang et al., 2022b): Leverages a pool of learnable prompts, dynamically retrieving the most relevant ones per input to adapt to new tasks without altering the backbone model.

- **LFPT5** (Huang et al., 2021): A continual learning variant of T5 that optimizes task-specific prompts and generates pseudo-data for rehearsal, combining prompting with generative memory.

- **IncLoRA**: Allocates one dedicated LoRA module per task and freezes previous adapters, enabling incremental adaptation while isolating task-specific knowledge.

- **MIGU** (Du et al., 2024): Introduces a magnitude-based gradient mask that updates only parameters with significant changes, assuming task-specific parameter importance distributions. We apply MIGU on top of IncLoRA in our baseline.

- **O-LoRA** (Wang et al., 2023b): Enhances IncLoRA by enforcing orthogonality between task-specific parameter updates, thereby reducing subspace interference across tasks.

- **LB-CL** (Qiao & Mahdavi, 2024): Builds upon IncLoRA by initializing new LoRA modules via SVD decomposition of previous ones and projecting gradients to orthogonal subspaces to separate task knowledge.

- **MoCL** (Wang et al., 2024a): Dynamically fuses previously trained LoRA modules based on computed task similarity scores, promoting knowledge reuse while minimizing interference.

## D  MORE RESULTS

**Detailed Comparison of DF-CL with Direct Adaptation of DSF.**  As previously discussed in Section 3.2, directly applying the Discrete Fourier Transform leads to temporal forgetting, illustrated in Figure 3. To address this issue, we introduced task-specific branches, which help reduce performance instability during continual training. However, residual forgetting remains, motivating the design of the task-weight merging strategy. To further validate its effectiveness, we conduct an ablation study and visualize the performance of MNLI and CB across the full training sequence in Figure 3. The results clearly show that our merging strategy effectively mitigates forgetting. The combination of task-specific branches and task-weight merging—constituting our proposed DF-CL framework—provides a robust solution for continual learning with improved stability and retention.

**Average Accuracy across Long Tasks.**  The design of DF-CL aims to ensure task stability across varying task orderings. To validate this, we further evaluate the average performance under different task sequences, as shown in Figure D1. Despite the variations in task order, DF-CL consistently maintains stable performance. This indicates that our method is not only effective in mitigating interference from dissimilar tasks but also demonstrates a notable degree of robustness to task order, which is critical in NLP continual learning settings.

**All Task Performance.**  We present the detailed performance comparison between O-LoRA and DF-CL on the Long Order 1 task sequence in Table D4. Notably, O-LoRA suffers from severe catastrophic forgetting on earlier tasks when trained on long task sequences. For example, as shown in Table X, the performance on the first task (MNLI) drops dramatically from 84.9 to 37.5 after training through task 15. In contrast, DF-CL maintains competitive performance, especially on early-stage tasks, demonstrating stronger resistance to forgetting. In addition, a comparison with other baselines across all 15 tasks is illustrated in Figure D2. DF-CL consistently performs competitively and nearly surpasses both O-LoRA and IncLoRA across all three task orderings. These results further highlight the generalization ability and stability of DF-CL, validating the effectiveness of our task-specific branches and task-weight merging strategy in mitigating forgetting across long task sequences.

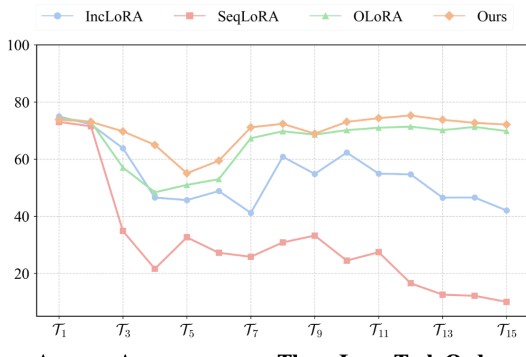

**Average Accuracy across Three Long Task Orders**

Figure D1: Average performance over three task orders on the Long Benchmark, reflecting the order sensitivity of different methods.

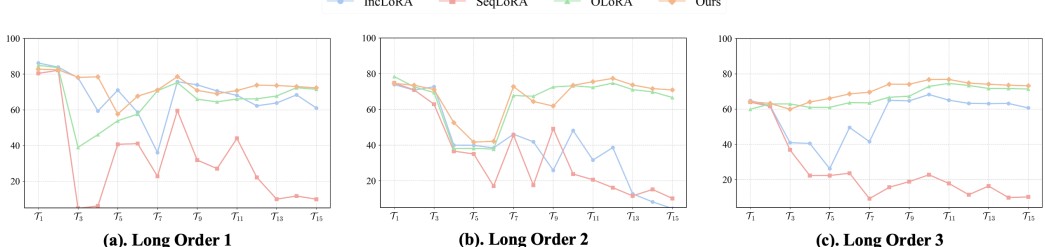

(a). Long Order 1          (b). Long Order 2          (c). Long Order 3

Figure D2: Detailed performance for each task on Long Benchmark with T5-Large model.

| O-Lora | MNLI | CB | WiC | COPA | QQP | BoolQA | RTE | IMDB | yelp | amazon | SST-2 | dbpedia | agnews | MultiRC | yahoo | avg |
|---|---|---|---|---|---|---|---|---|---|---|---|---|---|---|---|---|
| round1 | 84.9 | - | - | - | - | - | - | - | - | - | - | - | - | - | - | 84.9 |
| round2 | 83.6 | 89.3 | - | - | - | - | - | - | - | - | - | - | - | - | - | 83.6 |
| round3 | 37.4 | 32.1 | 57.8 | - | - | - | - | - | - | - | - | - | - | - | - | 39.0 |
| round4 | 45.5 | 35.7 | 54.2 | 48.0 | - | - | - | - | - | - | - | - | - | - | - | 46.1 |
| round5 | 26.3 | 17.9 | 52.0 | 40.0 | 82.0 | - | - | - | - | - | - | - | - | - | - | 53.9 |
| round6 | 29.8 | 21.4 | 58.0 | 53.0 | 75.9 | 80.2 | - | - | - | - | - | - | - | - | - | 57.6 |
| round7 | 61.9 | 76.8 | 58.9 | 50.0 | 75.9 | 80.8 | 83.4 | - | - | - | - | - | - | - | - | 70.7 |
| round8 | 62.4 | 76.8 | 50.5 | 48.0 | 72.1 | 75.9 | 80.5 | 93.2 | - | - | - | - | - | - | - | 75.3 |
| round9 | 56.2 | 67.9 | 50.2 | 58.0 | 55.1 | 70.3 | 78.3 | 91.9 | 59.7 | - | - | - | - | - | - | 66.0 |
| round10 | 54.2 | 67.9 | 50.2 | 56.0 | 53.5 | 71.4 | 76.9 | 91.2 | 62.6 | 58.5 | - | - | - | - | - | 64.4 |
| round11 | 50.9 | 60.7 | 52.0 | 49.0 | 57.5 | 74.6 | 70.4 | 93.6 | 63.6 | 59.2 | 93.8 | - | - | - | - | 66.1 |
| round12 | 39.4 | 50.0 | 51.6 | 55.0 | 46.3 | 69.1 | 52.7 | 94.4 | 59.2 | 56.3 | 93.7 | 98.6 | - | - | - | 66.1 |
| round13 | 37.7 | 50.0 | 50.6 | 62.0 | 41.2 | 69.0 | 57.8 | 94.3 | 58.4 | 55.4 | 93.1 | 98.2 | 87.5 | - | - | 67.7 |
| round14 | 37.4 | 50.0 | 53.9 | 64.0 | 76.0 | 78.2 | 53.8 | 94.4 | 56.8 | 54.3 | 93.0 | 98.2 | 87.4 | 71.8 | - | 72.4 |
| round15 | 37.5 | 50.0 | 50.8 | 56.0 | 75.4 | 74.8 | 54.2 | 94.4 | 56.1 | 54.0 | 91.7 | 98.1 | 85.2 | 69.9 | 71.1 | 71.5 |

| DF-CL | MNLI | CB | WiC | COPA | QQP | BoolQA | RTE | IMDB | yelp | amazon | SST-2 | dbpedia | agnews | MultiRC | yahoo | avg |
|---|---|---|---|---|---|---|---|---|---|---|---|---|---|---|---|---|
| round1 | 82.8 | - | - | - | - | - | - | - | - | - | - | - | - | - | - | 82.8 |
| round2 | 82.4 | 85.7 | - | - | - | - | - | - | - | - | - | - | - | - | - | 82.4 |
| round3 | 81.9 | 85.7 | 32.6 | - | - | - | - | - | - | - | - | - | - | - | - | 78.2 |
| round4 | 81.5 | 83.9 | 54.1 | 0.0 | - | - | - | - | - | - | - | - | - | - | - | 78.4 |
| round5 | 81.6 | 83.9 | 55.5 | 7.0 | 34.1 | - | - | - | - | - | - | - | - | - | - | 57.6 |
| round6 | 81.5 | 83.9 | 55.3 | 17.0 | 51.8 | 75.8 | - | - | - | - | - | - | - | - | - | 67.6 |
| round7 | 78.8 | 83.9 | 55.3 | 24.0 | 62.3 | 77.4 | 76.2 | - | - | - | - | - | - | - | - | 71.1 |
| round8 | 73.6 | 82.1 | 51.6 | 48.0 | 77.1 | 78.9 | 81.2 | 87.3 | - | - | - | - | - | - | - | 78.6 |
| round9 | 72.6 | 82.1 | 52.7 | 52.0 | 74.5 | 78.2 | 81.9 | 92.3 | 42.1 | - | - | - | - | - | - | 70.9 |
| round10 | 72.3 | 82.1 | 53.8 | 54.0 | 72.8 | 77.7 | 82.3 | 93.1 | 54.5 | 49.4 | - | - | - | - | - | 69.0 |
| round11 | 72.1 | 82.1 | 54.2 | 59.0 | 70.3 | 77.4 | 81.6 | 93.2 | 59.6 | 54.2 | 93.1 | - | - | - | - | 70.8 |
| round12 | 71.1 | 85.7 | 54.5 | 44.0 | 67.5 | 76.1 | 81.6 | 93.2 | 58.9 | 54.6 | 94.7 | 95.8 | - | - | - | 73.8 |
| round13 | 68.8 | 85.7 | 53.3 | 44.0 | 65.8 | 74.8 | 80.9 | 93.3 | 55.8 | 52.6 | 94.4 | 97.4 | 80.3 | - | - | 73.6 |
| round14 | 67.2 | 85.7 | 54.5 | 43.0 | 71.0 | 76.4 | 82.3 | 93.3 | 53.8 | 51.3 | 94.2 | 97.4 | 83.2 | 60.1 | - | 73.0 |
| round15 | 62.4 | 78.6 | 51.7 | 65.0 | 72.8 | 74.0 | 76.9 | 93.7 | 52.1 | 49.7 | 93.8 | 97.5 | 85.3 | 65.9 | 67.2 | 72.3 |

Table D4: Detailed comparison of Long1 results between O-LoRA and DF-CL. For each training round, the accuracy on all previously seen tasks is reported.

# E DISCUSSION

**Limitations.** While DF-CL demonstrates strong performance on a range of classification-focused NLP tasks, its applicability to more complex scenarios—such as reasoning-intensive tasks or open-ended generation—has not yet been explored. These tasks often involve richer contextual dependencies and are more sensitive to temporal forgetting, which may challenge the current spectral representation design.

**Future Work.** Future research can extend DF-CL to more challenging task types, including reasoning, structured prediction, and open-ended generation, to further assess its generalization capabilities. Another promising direction is to explore adaptive allocation of spectral coefficients—both globally and per task—based on task complexity or similarity. This could improve parameter efficiency and flexibility in highly heterogeneous task sequences. Additionally, enhancing the merging mechanism beyond magnitude-based selection—for example, via learned fusion, may lead to better handling of task-specific variation across tasks.

