# OpenReview forum: "Fourier Minds, Forget Less: Discrete Fourier Transform for Fast and Robust Continual Learning in LLMs"
_ICLR.cc/2026/Conference — ICLR 2026 Conference Withdrawn Submission_

### Official Review · Reviewer_Mi3V · 2025-10-31

**Soundness:** 2
**Presentation:** 2
**Contribution:** 2
**Rating:** 2
**Confidence:** 5

**Summary:**

This paper aims to address the cumulative parameter budget growing with the number of sequential tasks for continual learning in LLMs. Based on the instability and forgetting issue of directly applying Sparse Fourier Transform in continual learning, the authors propose Discrete Fourier Continual Learning (DF-CL), which decouples general and task-specific knowledge during training and utilizes a max-magnitude merging strategy in the post-training phase, to maintain the previous knowledge as well as adapt to new tasks. Experimental results show that DF-CL achieves superior performance and reduces the trainable parameters than other methods.

**Strengths:**

1. The problem this paper aims to address is important for continual learning in LLMs, since existing methods almost initialize a newly added LoRA for a new task, which heavily increase the cost with the growth of the sequential tasks.

2. The proposed DF-CL method can reduce the number of trainable parameters for sequential tasks.

**Weaknesses:**

1. The paper does not provide any theoretical analysis to support the principle of the proposed DF-CL, especially for the max-magnitude merging method. The paper does not clearly explain why DF-CL chooses this merging method theoretically.

2. The hypothesis of DF-CL proposed in this paper lacks support. In line 226, it claims the motivation and reason “we first hypothesize that the observed stability gap stems from the use of a shared coefficient vector x across all tasks” for utilizing task-specific branch in DF-CL, but the paper directly makes such statements without any analysis or related work. This makes the process of proposing task-specific coefficients not convincing.

3. The experiments only use classification tasks, which are relatively simple to evaluate the performance of the proposed DF-CL in LLMs. NLP Generation tasks like SuperNI benchmark have been utilized in parameter-efficient continual learning, but this paper does not conduct experiments on such benchmarks.

4. To show the performance of DF-CL, the paper tried to show the accuracy flow of the first task across sequential tasks, such as Figure 1(c). But this result cannot fully evaluate DF-CL, since the metric average accuracy reflects not only previous tasks' accuracy but also new task accuracy. If only showing the accuracy of the first task, especially in an order of four tasks, it cannot explain how the proposed method achieves the trade-off between maintaining previous knowledge and adapting to new tasks.

5. The evaluation metric is only the average accuracy. There are other common metrics, such as OPD [1], to better show the performance of the method.

[1] Scalable and Order-robust Continual Learning with Additive Parameter Decomposition, ICLR2020.

6. The experimental tables in this paper are not presented in an organized way, and also do not explain well for each table. For example, Figure 3 does not define the meaning of x-axis.

**Questions:**

1. During training, since DF-CL still freezes previous tasks' task-specific coefficient matrices, can we have the conclusion that DF-CL does not save computational memory during training? Since in the forward, previous task task-specific coefficient matrices still join the computation for the output.

2. DF-CL involves merging strategy after all tasks training. But the problem is, after addressing current sequential tasks, when another task comes, how does DF-CL balance this new task and the previously merged task-specific coefficient matrices? Why not change DF-CL to merge each task directly after its training? Why does DF-CL choose to merge all tasks coefficient matrices after all tasks training?

3. Can authors compare the experimental performance of DF-CL with that of SAPT [1] and InfLoRA [2]? Also, can authors use SuperNI benchmark to evaluate DF-CL?

[1] SAPT: A Shared Attention Framework for Parameter-Efficient Continual Learning of Large Language Models, ACL2024.

[2] InfLoRA: Interference-Free Low-Rank Adaptation for Continual Learning, CVPR2024.

4. Figure 1(c) only shows the performance of the first task drops down may not be solid enough to show the effectiveness of DF-CL, and that’s why the common metric is average accuracy.

5. Figure 3 does not define the meaning of x-axis. For readers who are not familiar with continual learning, it’s a little hard to understand. Figure 3 only lists the performance of the first task to support the claim “instability” of directly applying SFT, which is not solid.

6. There is no clear definition of “sensitivity of model weights to the tasks” and “instability of task performance” in the paper, which makes it confusing about how to distinguish them.

---

> ### Author Response · Authors · 2025-11-27
> **Response to Reviewer Mi3V's Questions**
>
> > We appreciate the reviewer's careful evaluation of our work. In the following, we respond to each comment in detail and clarify the points that were not sufficiently explained in the original submission.
>
> **Q1: Memory usage comparison**
> > - Thank you for the question. We clarify the memory behavior as follows. We agree that DF-CL must load previous task-specific coefficients during the forward pass. However, in large-scale LLM training, **backward pass and optimizer states dominate memory usage**, not the forward pass.
> > - Crucially, **only the current task’s coefficients are trainable**, while all previous task-specific branches are frozen. Thus, they require **no gradients, optimizer states, or momentum buffers**.
> >   As a result, the memory for the backward pass is greatly reduced: 19MB (O-LoRA) vs. DF-CL (1.2MB). This ~16× reduction directly lowers backpropagation and optimizer memory, which dominate total training memory in LLMs.
>
> **Q2: Clarification on the merging strategy**
> > - (**Progressive vs. final merging**): In our implementation, the **max-magnitude merging is applied progressively after each task**, not only once at the end. Since the operation is element-wise and both *commutative* and *associative*, $$max(a,b,c)=max(max(a,b),c),$$
> >   progressive merging is mathematically equivalent to merging all task-specific matrices at the end. We will clarify this in the revised manuscript.
> > - (**Why merging helps with new tasks**): After merging, the consolidated matrix serves as a **stable global representation shared across tasks**. When a new task arrives, only its own task-specific coefficients are updated; previously merged components remain fixed, preventing interference.
> > - (**Empirical justification**): As shown in Table 6, the max-magnitude strategy consistently improves both accuracy and forgetting metrics compared to the non-merged variant.
>
> **Q3: Missing baselines and benchmarks**
> > - Thank you for highlighting this point. We have incorporated additional comparisons with several recently proposed PEFT-based CL baselines.
> > - We evaluate SD-LoRA [1], SAPT, and InfLoRA on the Standard Benchmark using T5 models. The results are summarized below:
> >
> >   |Method|Short1|Short2|Short3|
> >   | -- | -- | -- | -- |
> >   | SD-LoRA|67.7|55.9| 60.9|
> >   | SAPT| 70.4|71.1|68.9|
> >   | InfLoRA| 75.7 |75.2| 72.8|
> >   | DF-CL| **78.7**|**78.7**|**78.4** |
> >
> >   DF-CL consistently outperforms these baselines with a clear margin.
> >   We emphasize that:
> >
> >   - **SAPT** requires higher memory due to its shared-attention routing mechanism.
> >   - **InfLoRA** incurs substantial computational overhead from gradient-space projection.
> >   - In contrast, **DF-CL maintains strong performance while remaining highly parameter-efficient**.
> >
> > - We further evaluate DF-CL on SuperNI, following two settings:
> >   - **SuperNI-G:** all generation tasks
> >   - **SuperNI-SG:** summarization and two-generation mixed tasks
> >
> >   | Method | SuperNI-G | SuperNI-SG |
> >   | --- | -- | -- |
> >   | O-LoRA | 12.13 | 34.22 |
> >   | SAPT | 12.66 | 34.48 |
> >   | DF-CL| **13.04**|**35.28**|
> >
> >   DF-CL achieves the best performance in both settings, demonstrating its effectiveness beyond classification tasks and showing strong generalization on generation-oriented evaluation.
>
> **Q4 & W4. The accuracy flow of the first task (Figure 1(c)) is not sufficient to support the effectiveness of the proposed method.**
> > - Figure 1(c) is intended only as a simple illustrative example using the first two tasks. For a complete evaluation, we report the **average accuracy across all tasks after each training step** in Figure 4 and Figure D1, which reveal the full performance trajectories and clearly show the stability–plasticity behavior of DF-CL compared with multiple baselines.
> > - In addition, Appendix Table D4 provides the **per-task performance changes across all 15 sequential tasks**, where DF-CL consistently exhibits much smaller fluctuations than O-LoRA, further demonstrating its improved stability over long task sequences.
>
> **Q5: Figure caption**
> > - Thank you for pointing this out. In Figure 3, the **x-axis represents the sequential task index**, and we will make this explicit in the revised caption.
>
> **Q6: Clarification of Definition**
> > - By “sensitivity of model weights,” we refer to the fact that **small changes in spectral coefficients can induce disproportionately large shifts in the spatial-domain weights after IDFT**.
> > - By “instability of task performance,” we mean the **fluctuations in accuracy on earlier tasks during sequential training**. The former is a property of the DFT parameterization, while the latter is its observable consequence in continual learning.
> > - We will revise our draft to clarify these definitions.
>
> [1]. SD-LoRA: Scalable Decoupled Low-Rank Adaptation for Class Incremental Learning. ICLR 2025.

---

> ### Author Response · Authors · 2025-11-27
> **Response to Reviewer Mi3V's Weaknesses**
>
> **W1. Analysis of merging strategy**
>
> > - Thank you for raising this point. While our method is primarily empirically driven, we clarify the motivation and supporting evidence below.
> >
> > - As shown in our ablations (Table below), the max-magnitude rule consistently outperforms mean merging across all task orders:
> >
> >   | Method | Order1   | Order2   | Order3   |
> >   | ------ | -------- | -------- | -------- |
> >   | Mean   | 76.8     | 64.1     | 77.8     |
> >   | Max    | **78.7** | **78.7** | **78.4** |
> >
> > - Higher-magnitude coefficients tend to encode more stable or repeatedly reinforced information, while smaller values are more likely to reflect task-specific noise. Thus, retaining the maximum magnitude across tasks helps preserve shared, high-saliency updates.
> >
> > - We agree that a full theoretical treatment of the merging rule would strengthen the framework, and we plan to explore this direction in future work.
>
> **W2.The hypothesis for introducing task-specific coefficients lacks support.**
>
> > - A single shared coefficient vector $x$ forces *all tasks* to update the same spectral parameters. Because small spectral changes translate into large weight changes after IDFT (as shown in Fig. 3(b)), updates from later tasks inevitably overwrite earlier ones. This creates the *stability gap* we observe in sequential training.
> > - Figure 3(a) shows that the naive, fully-shared DFT baseline exhibits pronounced performance drops on previous tasks during training. Appendix Table D4 further confirms this effect across all 15 tasks, where shared-only updates lead to large fluctuations. Introducing task-specific coefficients isolates these updates, and the instability disappears.
> > - Our design is also consistent with a long line of CL research [2-3] emphasizing the need to separate shared vs. task-specific parameters
>
> **W3. The experiments only use classification tasks.  NLP Generation tasks like SuperNI benchmark have been utilized in parameter-efficient continual learning, but this paper does not conduct experiments on such benchmarks.**
>
> > - As suggested, we additionally evaluate DF-CL on NLP generation tasks from the SuperNI benchmark. The new results are reported in the following table.
> >
> >   - **SuperNI-G:** all generation tasks
> >
> >   - **SuperNI-SG:** summarization and two-generation mixed tasks
> >
> >   | Method | SuperNI-G | SuperNI-SG |
> >   | ------ | --------- | ---------- |
> >   | O-LoRA | 12.13     | 34.22      |
> >   | SAPT   | 12.66     | 34.48      |
> >   | DF-CL  | **13.04** | **35.28**  |
> >
> > - It can be observed that DF-CL **consistently outperforms the baselines** and shows similar gains as in the classification benchmarks, suggesting that our approach generalizes beyond classification to sequence generation tasks.
>
> **W5. The evaluation metric is only the average accuracy.**
>
> > - Following the reviewer’s suggestion, we now report three additional CL metrics: **forgetting rate (FR), backward transfer (BWT), and forward transfer (FWT).** FR measures how much performance on past tasks degrades after learning new ones, BWT quantifies the influence of learning new tasks on previous tasks, and FWT reflects positive transfer to future tasks.
> >
> >   | T5 (Standard) | FR ↓ | FWT ↑ | BWT ↑ |
> >   | --- | ----- | ----- | -- |
> >   | SeqLoRA  | 67.82| -0.29 | -67.82|
> >   | IncLoRA | 5.07 | -0.46 | -5.00|
> >   | O-LoRA | 3.29 | -0.46 | -3.26 |
> >   | MoCL  | 4.14 | -2.37 | -2.11|
> >   | **DF-CL** | **0.84** | -0.43 | **-0.54** |
> >
> >   The results show that DF-CL achieves lower FR and more favorable BWT/FWT than the baselines, consistent with the improvements observed in average accuracy.
>
> [2]. Cs2K: Class-specific and Class-shared Knowledge Guidance for Incremental Semantic Segmentation. ECCV 2024.
>
> [3]. Task-aware Orthogonal Sparse Network for Exploring Shared Knowledge in Continual Learning. ICML 2024.

---

### Official Review · Reviewer_XmWV · 2025-11-01

**Soundness:** 3
**Presentation:** 3
**Contribution:** 2
**Rating:** 2
**Confidence:** 3

**Summary:**

This paper proposes a discrete Fourier continual learning algorithm, a continual learning framework for large language models based on the sparse Fourier Transform. The proposed method decomposes model updates into spectral components, separating shared and task-specific knowledge while enforcing orthogonality among tasks. It introduces a max-magnitude weight merging strategy to consolidate task-specific adaptations into the global model. Experiments on T5-Large model and LLaMA2-7B model across standard and long continual learning benchmarks show improvements over LoRA-based baselines with only 1–3% of trainable parameters.

**Strengths:**

1. This paper is well structured, provides sufficient implementation details, and includes explicit training configurations and baselines, supporting reproducibility of the proposed method.
2. As the authors claimed in Table 1, the proposed method greatly reduces trainable parameters without sacrificing accuracy, which is significant for scaling continual learning with large language models.

**Weaknesses:**

1. The idea of studying sparse Fourier Transform in continual learning is only superficially motivated. The authors claim that frequency decomposition separates “shared” and “task-specific” knowledge, but this analogy is not well justified. No analysis is given on what the frequency components represent in the model weights or how this relates to forgetting dynamics. The method appears as an arbitrary parameter reparameterization rather than a principled CL framework. To conclude, this is rather heuristic, while lack of theoretical gurantees.
2. The proposed approach mainly combines existing ideas of FourierFT parameterization and the LoRA-like task decomposition, without introducing a fundamentally new mechanism for mitigating forgetting in continual learning. The “max-magnitude merging” is a simple heuristic lacking theoretical analysis or ablation explaining why it should preserve knowledge.
3. The experiments focus exclusively on text classification datasets. No reasoning, generation, or open-ended tasks are evaluated, which makes the results less impactful for LLM continual learning.
4. Baselines are limited and not fairly tuned. Classical regularization-based methods are either omitted or weakly configured. And the reported improvements are small (often within 1–2%), and there is no statistical analysis to assess significance.
5. The paper is a little bit verbose.

**Questions:**

1. Will the method be instabile if tasks are highly correlated, e.g., sequential fine-tuning on similar domains?
2. Is the merging operation performed once at the end or progressively after each task?

---

> ### Author Response · Authors · 2025-11-27
> **Response to Reviewer XmWV**
>
> > Thanks for your comments. We will answer the question and discuss point by point as follows. We hope that our response satisfactorily addresses the issues you raised. Please feel free to let us know if you have any additional concerns or questions.
>
> **Q1&W3. The experiments focus exclusively on text classification datasets. No reasoning, generation, or open-ended tasks are evaluated.**
>
> > - Thank you for pointing this out. In addition to classification tasks, we have conducted experiments on **generation-oriented continual learning** using the SuperNI benchmark. We evaluate two settings:
> >
> >   - **SuperNI-G:** all generation tasks
> >   - **SuperNI-SG:** summarization and generation mixed tasks
> >
> > - The results are shown below:
> >
> >   | Method | SuperNI-G | SuperNI-SG |
> >   | ------ | --------- | ---------- |
> >   | O-LoRA | 12.13     | 34.22      |
> >   | SAPT   | 12.66     | 34.48      |
> >   | DF-CL  | **13.04** | **35.28**  |
> >
> >   DF-CL achieves consistently **better performance than SAPT and O-LoRA**, indicating that the proposed method generalizes beyond classification and remains effective on more **complex generation and summarization** tasks.
>
> **Q2. Is the merging operation performed once at the end or progressively after each task?**
>
> > - We merge task-specific components **progressively after each task** using the max-magnitude rule. Since the operation is element-wise and both **commutative and associative**,
> >
> >   $$max⁡(a,b,c)=max⁡(max⁡(a,b),c),max(a,b,c)=max(max(a,b),c),$$
> >
> >   progressive merging is mathematically equivalent to applying the merge once at the end. We will clarify this implementation detail in the revised version.
>
>
> **W1. On the motivation and principled grounding of using the sparse Fourier Transform for CL.**
>
> > - Our DF-CL is **not** based on interpreting specific frequencies as “shared” or “task‐specific.” Instead, it leverages two fundamental mathematical properties of Fourier representations:
> >
> >   - **Orthogonality:** Each frequency component corresponds to a basis vector orthogonal to all others.
> >     Updating disjoint frequency indices therefore produces weight updates in **orthogonal subspaces** in the spatial domain.
> >
> >   - **Sparsity:** Only a small number of coefficients need to be updated, making the parameterization naturally efficient.
> >
> > - These structural properties directly align with continual learning’s goal of **separating task-specific updates to reduce interference**.
> > - We empirically observe (Fig. 3b) that **small spectral perturbations translate into large spatial-domain weight changes**, making a fully shared spectral parameterization unstable. Allocating **disjoint frequency subsets to different tasks** isolates their updates into orthogonal subspaces, preventing new tasks from overriding previously learned information.
>
>
> **W2-1. The proposed approach mainly combines existing ideas of FourierFT parameterization and the LoRA-like task decomposition, without introducing a fundamentally new mechanism for mitigating forgetting in continual learning.**
>
> > We appreciate the reviewer's comment. We are not simply combining Fourier-based parameterization with a LoRA-like decomposition, but tailoring FourierFT specifically to the efficiency and scalability bottlenecks of continual learning.
> >
> > - In long task sequences, CL methods must repeatedly adapt to new tasks under tight memory and computation budgets; the **FourierFT parameterization** enables us to represent and update model weights in a sparse frequency domain, which **substantially improves parameter- and compute-efficiency for continual updates**.
> > - A naive application of FourierFT to CL does not directly work in CL scenarios. The key contribution of our method is the **explicit design of shared and task-specific components in the Fourier domain, together with a model-merging mechanism** that allows us to accumulate knowledge across tasks without retraining from scratch.

---

> ### Author Response · Authors · 2025-11-27
> **Response to Reviewer XmWV (part II)**
>
> **W2-2. The “max-magnitude merging” is a simple heuristic lacking theoretical analysis or ablation explaining why it should preserve knowledge.**
>
> > - While max-magnitude merging is indeed a heuristic, we empirically verify its effectiveness.
> >
> >   | Method    | Order1   | Order2   | Order3   |
> >   | --------- | -------- | -------- | -------- |
> >   | w/o Merge | 77.7     | 78.2     | 77.3     |
> >   | Mean      | 76.8     | 64.1     | 77.8     |
> >   | Max       | **78.7** | **78.7** | **78.4** |
> >
> > - As shown above, comparing our method with and without the merging step, the max-magnitude strategy consistently yields higher average accuracy and lower forgetting.
> > - Intuitively, similar to magnitude-based consolidation strategies in CL, larger-magnitude coefficients in the Fourier domain tend to encode more stable and repeatedly reinforced information, so keeping the maximum across tasks biases the merged model toward preserving such shared knowledge rather than task-specific noise.
>
> **W4. Baselines are limited and not fairly tuned.  Classical regularization-based methods are either omitted or weakly configured.**
>
> > As suggested, we did experiment with classical regularization-based baselines, including EWC[1] and LwF[2].
> >
> >   | T5 (Standard) | Order1 | Order2 | Order3 |
> >   | ------------- | ------ | ------ | ------ |
> >   | EWC           | 24.1   | 22.8   | 21.2   |
> >   | LwF           | 23.8   | 22.0   | 19.0   |
> >   | DF-CL         | **78.7**   | **78.7**   | **78.4**  |
> >
> > In standard benchmarks, their performance was very close to naive finetuning, and they almost completely forgot previous tasks. This behavior is consistent with the observations reported in [3].
>
> [1]. Overcoming catastrophic forgetting in neural networks. NeurIPS 2017.
>
> [2]. Learning without Forgetting. PANS 2017.
>
> [3]. Mitigating Catastrophic Forgetting in Online Continual Learning by Modeling Previous Task Interrelations via Pareto Optimization. ICML 2024.

---

### Official Review · Reviewer_ccLY · 2025-11-01

**Soundness:** 2
**Presentation:** 3
**Contribution:** 3
**Rating:** 6
**Confidence:** 4

**Summary:**

This paper proposes Discrete Fourier Continual Learning (DF-CL) for continual learning, which leverages Sparse Fourier Transform (SFT) for parameter efficiency. Experiments on T5 and LLaMA show DF-CL outperforms baselines.

**Strengths:**

1. This is the first work to apply SFT to CL, moving beyond LoRA/prompts to decouple knowledge via spectral properties.

2. Extreme parameter efficiency: DF-CL uses far fewer parameters (e.g., 120k vs. 11.8M for O-LoRA on T5-Large with 15 tasks), with the gap widening for more tasks/larger models.

**Weaknesses:**

1. Only evaluates classification tasks (sentiment, NLI, QA), not complex tasks like reasoning or open-ended generation.

2. Claims Fourier bases’ orthogonality reduces interference but lacks formal proof or quantitative overlap analysis.

3. Focuses on parameter efficiency but not IDFT’s computational cost for large weight matrices (e.g., LLaMA2-7B’s 4096×4096).

**Questions:**

1. Will DF-CL work for non-classification tasks (e.g., generation)?

2. How is Fourier bases’ orthogonality preserved during training (e.g., no global coefficient leakage into task subspaces)?

3. What’s DF-CL’s training latency vs. O-LoRA, especially IDFT’s cost?

---

> ### Author Response · Authors · 2025-11-27
> **Response to Reviewer ccLY**
>
> > Your constructive comments on our paper are highly valued. Please find our point-by-point response below. We hope this addresses all the issues you raised. Please let us know if you have any further questions.
>
> **W1&Q1: There is no complex task like reasoning or open-ended generation.**
>
> > - We appreciate your suggestion. To ensure a direct comparison with current SOTA algorithms (**O-LoRA, LB-CL and MoCL**), we initially followed the established benchmarks, which mainly focus on classification tasks (sentiment, NLI, QA).
> >
> > - To address your concern and further validate the robustness of our method, we are currently performing experiments on generation tasks using the SuperNI benchmark. The setup for the generation tasks is as follows:
> >
> >   - **SuperNI-G:** all generation tasks
> >   - **SuperNI-SG:** summarization and generation mixed tasks
> >
> >   | Method | SuperNI-G $\uparrow$ | SuperNI-SG  $\uparrow$ |
> >   | ------ | --------- | ---------- |
> >   | O-LoRA | 12.13     | 34.22      |
> >   | SAPT   | 12.66     | 34.48      |
> >   | DF-CL  | **13.04** | **35.28**  |
> >
> >   Our method demonstrates a clear performance improvement compared to SAPT, even on these complex generation tasks, which further confirms the effectiveness of our algorithm.
>
>
> **W2: Claims Fourier bases’ orthogonality reduces interference but lacks formal proof or quantitative overlap analysis.**
>
> > Our core idea is to leverage the intrinsic orthogonality of the Fourier bases to achieve task decoupling, thereby reducing interference and catastrophic forgetting.
> >
> > - (**Mechanism Explained**). In our expandable framework, we assign an independent and orthogonal subset of frequencies as the basis for each task's LoRA weight update. Because the Fourier bases are mutually orthogonal, one task's weight update primarily affects only its corresponding frequency subset.
> > - (**Reducing Forgetting**). This design ensures that knowledge for each task is encoded in nearly non-overlapping regions of the parameter space. Consequently, when the model learns a new task, it mainly modifies the parameters corresponding to the new task's orthogonal basis, thereby minimizing interference with previously learned parameters and significantly reducing catastrophic forgetting.
> > - (**Quantitative Evidence**).To quantitatively prove this decoupling effect, we compare the performance change of task 1 across sequential training and find that our design decoupled signficianly slove the temporal forgetting problem.
>
>
> **W3&Q3.Benchmark IDFT's computational cost/latency on large matrices and compare DF-CL's training latency against O-LoRA.**
>
> > - Thank you for raising this important point. Although DF-CL introduces an IDFT operation during the forward pass, its computational overhead is small compared to the overall Transformer computation.
> > - To directly address the reviewer’s concern, we benchmark the **actual training latency** of DF-CL versus O-LoRA on the DBpedia task using a single RTX 4090 GPU. The results are: **778s (O-LoRA)** vs. **602s (DF-CL)**.
> > - Despite the IDFT operation, **DF-CL is faster** in practice. This is because DF-CL updates only a **very small number of spectral coefficients**, whereas O-LoRA optimizes significantly larger low-rank matrices across all layers. The reduced parameter size leads to lower optimizer cost and faster training iterations, outweighing the minor overhead of the IDFT.
> > - These empirical results demonstrate that DF-CL introduces **no practical latency drawback** and can even offer **lower training time** compared to strong LoRA-based CL baselines.
>
> **Q2.How is Fourier bases’ orthogonality preserved during training (e.g., no global coefficient leakage into task subspaces)?**
>
> > We thank the reviewer for raising this critical question regarding the robustness of our algorithm. Our architecture is designed to combine a Shared Basis with Task-Specific Fourier Bases to learn general knowledge and isolate task-specific information, respectively.
> >
> > - (**The Shared Component.**) This part of the parameters is intended to learn **common, cross-task features**, and thus is intentionally shared and updated by all tasks. "Leakage" or overlap here is expected, as it represents the integration of general knowledge.
> > - **The Task-Specific Fourier Component (Preserving Orthogonality)**. Orthogonality is preserved by assigning **disjoint frequency index sets** to different tasks. Since each task updates only the coefficients corresponding to its own index subset, the underlying Fourier bases remain orthogonal by construction.
> >
> > In short, by employing a strict sparse update strategy at the implementation level, we ensure that task-specific knowledge is confined to its proprietary orthogonal Fourier subspace, effectively preventing coefficient leakage between task-specific components and maintaining the isolation advantage provided by orthogonality.

---

### Official Review · Reviewer_zReZ · 2025-11-06

**Soundness:** 3
**Presentation:** 3
**Contribution:** 2
**Rating:** 4
**Confidence:** 4

**Summary:**

This paper proposes Discrete Fourier Continual Learning (DF-CL), a parameter-efficient method using the Sparse Fourier Transform (SFT) to separate shared and task-specific knowledge in continual learning. By leveraging Fourier orthogonality and a max-magnitude merging strategy, DF-CL reduces interference and forgetting, achieving strong performance on T5-Large and LLaMA2-7B with only 1–3% trainable parameters.

**Strengths:**

1. The paper is well-organized and easy to follow.

2. The investigated problem of continual learning using low-rank adaptation is important.

**Weaknesses:**

1. The paper appears to overlook several relevant studies on continual learning (or continual fine-tuning) of large language models, such as LoRA-MoE and TreeLoRA. Providing a discussion on how the proposed approach relates to or differs from these methods would help clarify its novelty and contribution.

   *LoRA-MoE: Alleviating World Knowledge Forgetting in Large Language Models via MoE-Style Plugin*

   *TreeLoRA: Efficient Continual Learning via Layer-Wise LoRAs Guided by a Hierarchical Gradient-Similarity Tree*

2. It would be appreciated if the authors could further elaborate on the main contributions of the paper. In particular, a clearer explanation of the fundamental challenge in integrating the Discrete Fourier Transformation with LoRA would strengthen the clarity and significance of the proposed method.

3. In the experimental section, it is unclear why only accuracy results are reported. Including the forgetting metrics and the standard deviation of results would provide a more comprehensive and reliable evaluation.

**Questions:**

See Weaknesses above.

---

> ### Author Response · Authors · 2025-11-27
> **Response to Reviewer zReZ**
>
> > Thanks for your time in dealing with our work. We will answer the question and discuss point by point as follows. We hope that our response satisfactorily addresses the issues you raised.
>
> **W1: Lack of discussion with relevant studies**
> > - Thank you for pointing out these relevant works. We have added detailed discussions comparing DF-CL with **TreeLoRA** and **LoRA-MoE** to clarify the distinctions and contributions.
> >   - **TreeLoRA** is an elegant and very interesting approach that constructs a layer-wise hierarchical tree of LoRA branches guided by gradient similarity. However, this approach relies on accurate gradient-based task relationships, which may become unreliable when task sequences are short, typical conditions in standard CL benchmarks. In contrast, **DF-CL does not depend on task similarity heuristics**. It directly separates global and task-specific knowledge in the spectral domain and enforces orthogonality via disjoint coefficient indices, enabling stable performance on both short and long task sequences.
> >   - **LoRA-MoE** mitigates knowledge forgetting through MoE-style routing, but requires maintaining multiple LoRA experts and incurs additional routing and inference overhead. DF-CL remains **much more lightweight**, using only one global spectral branch plus small task-specific coefficients with **constant inference cost** and **substantially higher parameter efficiency**.
> >
> > - We further reproduce TreeLoRA on the Standard benchmark (with 4 sequential tasks) using T5-Large:
> >
> >   | T5 | Order1 | Order2 | Order3 |
> >   | ------------- | ------ | ------ | ------ |
> >   |SeqLoRA |25.7| 24.0| 35.2|
> >   | TreeLoRA| 44.0   | 52.8   | 44.0   |
> >   | DF-CL | 78.7   | 78.7   | 78.4   |
> >
> >   TreeLoRA generally outperforms simpler LoRA-based CL methods such as SeqLoRA, demonstrating its effectiveness. However, under the same Standard benchmark setting, DF-CL still achieves higher performance, likely because TreeLoRA’s KD-tree routing is unreliable with only a few tasks. In contrast, DF-CL does not rely on task similarity heuristics, enabling more stable and consistent performance.
> >
> > - Due to time constraints, we report TreeLoRA results only on the Standard benchmark. If you have specific concerns or preferred settings, we would be glad to run those as well.
>
> **W2: Main contributions and the explanation of the fundamental challenge in integrating the Discrete Fourier Transformation with LoRA**
> > - Directly applying DFT within LoRA leads to severe instability in CL: small spectral updates cause large, non-local changes to the weight matrix, resulting in strong task interference. Empirically, naive DFT+LoRA causes the first-task performance to drop from **79.8 → 52.1**, showing that the integration is non-trivial and requires CL-specific design.
> > - We also summarize our **Main contributions** as follows:
> >   - **First exploration of DFT-based representations in continual learning**, offering a new direction beyond low-rank or prompt-based PEFT.
> >   - **A decoupled spectral architecture** with global and task-specific branches, using disjoint coefficient indices to prevent cross-task interference and eliminate temporal forgetting.
> >   - **A task-weight merging strategy** that preserves important weight changes and further stabilizes training.
>
> **W3: Missing Forgetting Metrics and Standard Deviation**
> > - Thank you for pointing this out. We fully agree that forgetting metrics and variance are essential for a comprehensive CL evaluation.
> > - We have now computed the average standard CL metrics - FR, BWT, and FWT - on the **T5 Standard benchmark** and added multi-seed results.
> >
> >   | T5 (Standard) | FR ↓ | FWT ↑ | BWT ↑ |
> >   | --- | ----- | ----- | -- |
> >   | SeqLoRA  | 67.82| -0.29 | -67.82|
> >   | IncLoRA | 5.07 | -0.46 | -5.00|
> >   | O-LoRA | 3.29 | -0.46 | -3.26 |
> >   | MoCL  | 4.14 | -2.37 | -2.11|
> >   | **DF-CL** | **0.84** | -0.43 | **-0.54** |
> >
> >   DF-CL obtains the **lowest forgetting (FR)** and **best backward transfer (BWT)** among all baselines, while keeping competitive forward transfer. This provides strong evidence that DF-CL substantially mitigates catastrophic forgetting.
> > - In the original submission, we followed the standard practice of prior LLM-based CL works such as O-LoRA and MoCL, which typically report single-run results. To address the reviewer’s concern, we now additionally report results under **three random seeds** on the Standard benchmark (T5):
> >
> >   | T5 (Standard) | Order1 | Order2  | Order3   |
> >   | -------- | ----- | ------- | ------ |
> >   | O-LoRA | 74.9$\pm$0.1 | 73.4$\pm$0.3 | 75.5$\pm$0.2 |
> >   | DF-CL | 78.6$\pm$0.1 | 78.6$\pm$0.1 | 78.5$\pm$0.4 |
> >
> >   DF-CL consistently outperforms O-LoRA by a clear margin across all orders, and the improvements are substantially larger than the observed standard deviations (≤0.4). Combined with the gains shown in Table 2, these results indicate that DF-CL’s improvements are **robust and statistically meaningful**.

---

### Note · Authors · 2025-12-15

I have read and agree with the venue's withdrawal policy on behalf of myself and my co-authors.